# Perceptually Visible but Emotionally Subliminal Stimuli to Improve Exposure Therapies

**DOI:** 10.3390/brainsci12070867

**Published:** 2022-06-30

**Authors:** Sergio Frumento, Angelo Gemignani, Danilo Menicucci

**Affiliations:** 1Department of Surgical, Medical, Molecular and Critical Area Pathology, University of Pisa, 56126 Pisa, Italy; sergiofrumento@gmail.com (S.F.); angelo.gemignani@unipi.it (A.G.); 2Clinical Psychology Branch, Azienda Ospedaliero-Universitaria Pisana, 56126 Pisa, Italy

**Keywords:** subliminal, unconscious, subthreshold, fear, phobia, desensitization, exposure therapy

## Abstract

Subliminal stimuli are gaining growing interest due to their capability to induce desensitization to pathologically feared (e.g., phobic) pictures without inducing exaggerated emotional reactions. However, unresolved methodological issues cast significant doubt on the reliability of these findings and their interpretation. The studies most robustly assessing stimulus detection found that ~30% of the supposed-to-be-subliminal stimuli were, in fact, detected, suggesting that the beneficial effects attributed to subliminal stimuli may result from those actually seen. Nevertheless, a deeper analysis of the data underlying this misinterpretation unveils theoretical and clinical implications. Since the purpose of subliminal stimulation is to reduce the aversiveness of exposure therapies while maintaining their efficacy, researchers should measure the emotional relevance of supposed-to-be-subliminal stimuli that are, in fact, detected. A distinction is needed between perceptually- and emotionally-subliminal stimuli: the former is not consciously detected; the latter just fails to elicit emotional reactions. Emotionally-subliminal stimuli could represent an intermediate step of exposure in addition to those involving perceptually subliminal or supraliminal stimuli. Importantly, emotionally subliminal stimuli could make patients able to sustain a conscious exposure to feared stimuli without exaggeratedly reacting to them: if confirmed by empirical data, this unexpected disconfirmation of patients’ beliefs could pave the way for successful therapy while increasing their self-efficacy and compliance to treatment.

## 1. Putting the Mental Back in Exposure Therapies

Can we reduce fear by merely extinguishing its physiological outputs? Researchers investigating emotions should cautiously infer such an inseparable association between subjective states and their physiological correlates, Taschereau-Dumouchel and colleagues claimed [1]. The authors pointed out that “treatments developed using more objective symptoms as a marker of psychopathology have mostly been disappointing in effectiveness”, inviting future researchers to base new therapeutic protocols on subjective measures (e.g., self-reports) [1] or on their specific psychophysiological correlates [2].

Here, we collect this invitation by introducing an emotion-centered classification of subliminal stimuli to «put the “mental” back» in the scientific debate concerning the therapeutic efficacy of subliminal exposure therapies: in our opinion, the issue raised by Taschereau-Dumouchel and colleagues [1] cannot leave this emerging research line out of consideration.

The debate dates back to the seminal study by Öhman & Soares [3], demonstrating that subliminal phobic stimuli elicit physiological correlates of phobic reaction (e.g., electrodermal reactions) in the absence of the subjective feeling of fear. Despite their efficacy, traditional exposure therapies are hardly accepted by the majority of phobic patients, and their clinical success is significantly affected by motivational factors [4]. Thus, the possibility of obtaining a comparable reduction of phobic symptoms through a safer, subliminal exposure gained a growing interest [5,6]. The observation that these protocols subliminally induced habituation in physiological reactions to phobic stimuli led some researchers [7] to incautiously compare their efficacy to that of traditional exposure therapy, despite the most clinically-relevant component of phobia (i.e., the subjective feeling of fear) was unaffected [6]. The root causes of this paradox (i.e., considering a phobia successfully treated even if phobic stimuli still induce subjective fear) lie in the inappropriate overlap between subjective emotions and their physiological and/or behavioral correlates [8,9]. In addition to the terminological confusion between these distinct concepts [9], the original definition of “subliminal pictures” as stimuli that are perceptually undetectable and unrecognizable [3] set unnecessary limitations to their therapeutic use. While the detectability criterion is fundamental in perception research, to “maintain the efficacy of exposure but reduce its aversiveness” [10] is more relevant for the exposed stimuli to be emotionally irrelevant rather than perceptually unseen. The definition of subliminal stimuli should adapt to include the possibility that stimuli supposed to be subliminal can fail to induce the expected subjective emotion even when they are, in fact, detected. While this possibility was never directly tested by any study, a deeper analysis of the scientific literature suggests that this hypothesis is already beyond mere speculation.

## 2. Introducing Emotionally-Subliminal Stimuli

Can a phobic stimulus fail to elicit fear thanks to its reduced duration, even if consciously detected? From the systematic review of the studies on subliminal phobic stimulations [6], it is reasonable to derive this conclusion. In fact, studies adopting a trial-by-trial assessment of stimulus awareness found that stimuli supposed to be subliminal were occasionally perceived: as an example, Sebastiani and colleagues excluded 27.7% (10/36) of participants because they detected several stimuli supposed to be subliminal lasting 20 ms [11]. These numbers highlight the relevance of performing a trial-by-trial check of stimulus detection, the neuroimaging correlates of which could represent a potential marker of neuropsychiatric disorders (as already reported in response to supraliminal stimuli [12]). Nevertheless, most of the reviewed studies failed to assess the effectiveness of masking on a trial-by-trial basis, making it hard to discern whether the results reported should be ascribed to the processing of unperceived stimuli rather than to the processing of the few stimuli possibly perceived despite the masking technique [6]. Ignoring this evidence, authoritative authors [10,13] keep considering as “subliminal” even phobic stimuli that are merely “very briefly” exposed (i.e., lasting 30 ms, as compared to an average of 20.5 ms adopted by different authors in 17 comparable studies [6]). Even in the absence of a trial-by-trial assessment of stimulus detection, the observed habituation effects were described as the result of “masked” phobic stimuli [13], allowing an “unconscious emotional learning” [10].

The concerns we have raised do not diminish the relevance of these results: rather, we think that a more rigorous interpretation could increase their impact. Indeed, the same authors also assessed the fear ratings following each block of stimuli: they found no significant increase in fear in phobic participants briefly exposed to phobic stimuli [13]. Given the low probability that every stimulus was successfully masked, we should consider the possibility that even phobic stimuli that were consciously detected were too brief to induce the conscious feeling of fear. This evidence would fit with our idea that the nervous circuits involved in detecting and fearing a phobic stimulus are different and should be integrated into a therapeutic protocol targeting both top-down and bottom-up processes involved in desensitization [5,6]. This could finally affect the “feeling a feeling” level described by Damasio [14] that could be erroneously thought to overlap perfectly with the subjective threshold of perception (i.e., the moment at which subjects can first discriminate stimuli, even if at a chance level [15]). The observation that very brief phobic stimuli can be detected without being feared is only compatible with the hypothesis that the emotional reaction (intended as the moment at which subjects can verbalize feeling fear) does not overlap with the ability of subjects to detect the stimulus, i.e., that perceptual threshold slightly precedes the emotional threshold (Figure 1). This hypothesis is consistent with the use of mental imagery as an early step of systematic desensitization, based on the assumption that merely thinking to feared stimuli represents a perceptually and emotionally weaker version of its clearly-visible exposure [16]—despite the significant differences in the networks activated by these exposure paradigms [17]. Indeed, fMRI correlates of the amygdala and inferotemporal visual cortex covaried in response to emotional picture exposure, linearly increasing with stimulus’ emotional salience [18]: the coupling between the conscious detection of a stimulus and its emotional (subjective and/or psychophysiological) correlates should be considered as a continuum.

Despite this evidence, there are currently no terminological differences to distinguish the reasons for which stimuli can fail to elicit the expected emotional response (i.e., missed detection or very-brief duration). To avoid further misunderstandings adding to those that already affect this research line [9,19], the introduction of a new category of “subliminal stimuli” is needed, distinguishing between “perceptually-subliminal” and “emotionally-subliminal” stimuli: the former corresponds to the traditional meaning of “subliminal” (i.e., consciously undetected, unreportable); the latter indicates emotional (e.g., fearful) stimuli that do not trigger the corresponding conscious feeling (e.g., fear) even if consciously perceived. In our opinion, this distinction—apparently negligible but actually essential—would allow therapists to find the perfect balance between treatment’s efficacy and acceptability: efficacy is best reached after a conscious exposure to phobic stimuli as subliminal exposure alone does not affect subjective fear [6]; acceptability is best reached if these stimuli are very brief (but not necessarily unseen).

Rather than perceptually-subliminal stimuli proposed by other authors [10], we propose that emotionally-subliminal stimuli could represent the best compromise between the acceptability and efficacy of exposure therapies (Figure 1).

## 3. How Emotionally-Subliminal Stimuli Could Improve Our Understanding and Treatment of Pathological Fear

Even if we hypothesize that exposure therapies based on emotionally-subliminal stimuli are less effective than current ones, their wider acceptability would still result in a higher absolute number of patients successfully treated. As an example, if emotionally-subliminal exposure therapies are proven to reach half the efficacy of current ones, they could overcompensate by reaching more than double the number of patients.

However, this is not the only nor the main reason to include emotionally-subliminal stimuli in exposure therapies, as the advantages and disadvantages mentioned so far are partially shared by perceptually-subliminal stimuli. Indeed, perceptually-subliminal stimuli too make exposure therapies more acceptable; they were demonstrated to successfully induce habituation in psychophysiological and behavioral correlates of phobic fear [21,22], definitely suppressing these symptoms rather than making them dormant (which would expose them to higher risks of relapses [23], in particular after adverse life events [24]). Despite these promising results, perceptually-subliminal stimuli were ineffective in reducing the symptom most relevant to patients, i.e., subjective fear [6]. This incongruency between the effects of subliminal exposure on fear (unaffected) and its correlates (successfully habituated) was hypothesized to rely on the inability of patients to shape their cognitive schemata after an improvement they are not aware of: if I’m describing myself since decades as a phobic person, the mere memory of being phobic will elicit a subjective feeling of fear—even in the absence of its correlates [6]. The first countermeasure to this phenomenon could be obtained by making the patient aware of the subliminally-induced habituation in psychophysiological correlates of fear: if I’m informed of such effect, it will be easier to interpret it in terms of a reduction in subjective fear [6]. As an example, if I realize that I’m no more shaking in front of a phobic stimulus, I could shape my cognitive schemata to think that my fear of it has vanished. This is consistent with the observation that perceptually-subliminal stimuli only reduced subjective fear when its self-report assessment was preceded by a supraliminal confrontation with the phobic stimulus [25,26]. It is possible to interpret this as the result of a cognitive remodeling that could only occur after a conscious appraisal of the phobic stimulus [6].

Based on a bifactorial model of fear, we proposed a double-step procedure exposing patients to perceptually subliminal and then to clearly-visible stimuli: this would impact both the cognitive and the defensive survival circuits, finally reducing the conscious feeling of fear and its psychophysiological correlates [6]. The desensitization coming from the exposure to either perceptually subliminal or clearly-visible stimuli comes with specular limitations: the former can be meaningless to patients as long as the cognitive circuit alone can trigger an emotional reaction to the feared stimulus; the latter directly reduces subjective fear and typically reduces psychophysiological correlates [27], but it could fail to entirely suppress the over-reactivity of the defensive survival circuit—thus exposing patients to a higher risk of relapses [6,23]. Poor connectivity between cognitive and defensive survival circuits could underlie this lack of mutual influence between subliminal and supraliminal exposure protocols: in fact, on what grounds should cognition be involved in the processing of stimuli that are not detected?

An emotionally-subliminal exposure could overcome these limitations by decoupling the perception of a feared stimulus from the expected emotional reaction: this could simultaneously involve both the cognitive and the defensive survival circuits, thus enhancing their connectivity and facilitating a mutual influence. In fact, the everyday-life experience of a phobic patient is that the perception of a phobic stimulus is immediately and inevitably followed by an overwhelming emotional reaction. Conversely, a deeper analysis of experimental methodologies and the related results suggest that perceiving and fearing a stimulus are two distinguishable processes, the effects of which emerge to consciousness at slightly shifted moments. Despite being small in timings, this decoupling could have relevant theoretical and clinical implications.

From a theoretical point of view, the existence of such a decoupling—between perceiving a fearful stimulus and being afraid of it—could support theories claiming that emotions arise as cognitive interpretations of physiological reactions, i.e., that the two processes (perception and emotion) are consecutive rather than parallel [20,28]. Such association could be necessary to establish an emotional valence to stimuli through classical conditioning [29]: if I shake the first time I see a spider, I could interpret it as being spider-fearful. However, evidence that fear persists even after successfully inducing habituation in physiological correlates suggests that the cognitive mechanism behind emotions does not necessarily need a physiological trigger to be activated (at least once the association has been previously established): if I’m used to being scared of spiders and describe myself as suffering from arachnophobia, physiological habituation to phobic stimuli could not be enough to suppress fear.

From a clinical point of view (as illustrated in Figure 2), ignoring the decoupling between perceiving and fearing an aversive stimulus could prevent exposure therapies from successfully breaking the stimulus-emotion bond established through fear conditioning. In this perspective, emotionally-relevant stimuli could be better than both alternatives (perceptually subliminal and clearly visible stimuli) at reaching the disconfirmation of beliefs and the violation of expectations behind the therapeutic success of desensitization protocols [30]. In fact, realizing that a brief—but, importantly, conscious—exposure to threats induces little to no fear could result in habituation of psychophysiological correlates going hand to hand with a therapeutic reshape of the maladaptive cognitive schemata underlying phobic symptomatology. This unexpected ability to sustain a supraliminal—even if very brief—exposure to phobic stimuli could improve a patient’s self-esteem, empowerment, and compliance to treatment, thus increasing the probability of a successful therapy [4].

These hypotheses require some empirical confirmation to be considered reliable and should be cautiously considered until a dedicated experiment will specifically test our theory. However, the present theoretical note lays the conceptual and terminological foundations on which future studies could justify their rationale. Furthermore, if the methodological concerns we raised about the scientific literature are consistent (i.e., if authors claiming to have administered perceptually-subliminal stimuli actually administered also emotionally-subliminal ones), the data coming from previous studies could represent a first experimental clue in favor of our theory. Future studies referring to—and empirically check-proofing—the conceptual framework proposed in Figure 2 could improve both reliability and clinical impact of the results coming from this research line.

## 4. Conclusions

Methodological concerns in the research line investigating the effects of subliminal exposure to phobic clues push for introducing an emotional threshold in stimulus processing. We propose to define a stimulus as emotionally subliminal if it does not induce—even if consciously perceived—the expected emotional reaction. As the observation that even unconscious stimuli induced psychophysiological correlates of fear [3] gave rise to a relevant scientific debate [6], the observation that even conscious stimuli can fail to induce subjective fear—if confirmed by dedicated experiments—could represent a turning point for the future of this research line.

Indeed, distinguishing between perceptually- and emotionally subliminal stimuli could significantly improve the understanding of emotions as well as the therapeutic protocols used to treat pathological fear.

## Figures and Tables

**Figure 1 brainsci-12-00867-f001:**
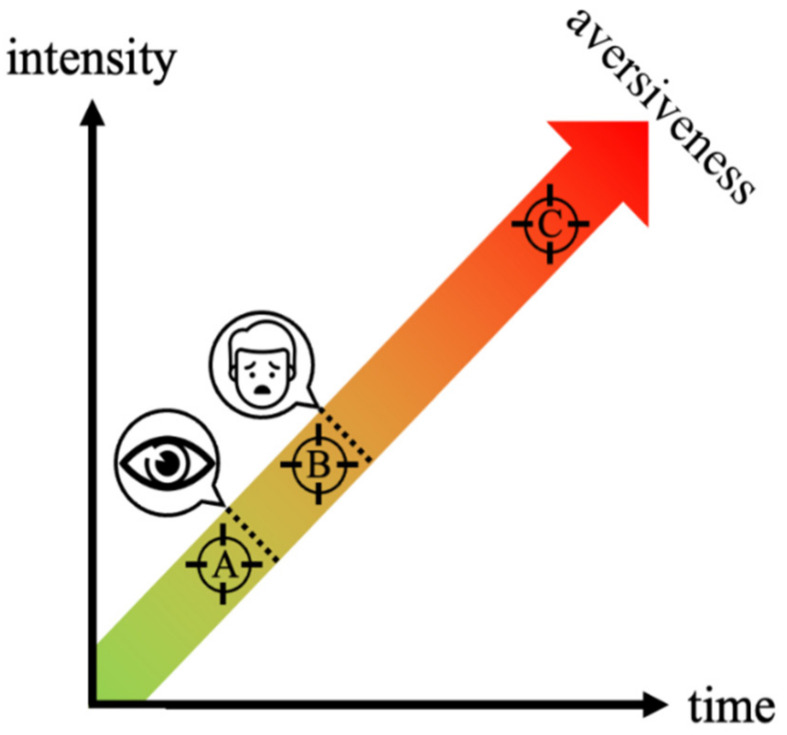
Time duration and intensity (perceptual and/or emotional) of phobic stimuli affect their emergence to consciousness and the acceptability of their exposure. Some authors [10,20] propose to shift the target of exposure therapies from clearly-visible and aversive stimuli (C) to perceptually-subliminal ones (A) to increase access and adherence to treatment. Based on studies assessing stimulus visibility trial-by-trial [6], it seems unlikely that all very brief (e.g., 30 ms [13]) stimuli were successfully masked (i.e., perceptually-subliminal); however, even those that were actually detected were still unable to elicit the expected fear reaction. This mismatch between the detection of phobic stimuli and the onset of the related emotional reaction would make these stimuli emotionally subliminal (B): this new class of stimuli could represent the best compromise between acceptability and efficacy of the exposure.

**Figure 2 brainsci-12-00867-f002:**
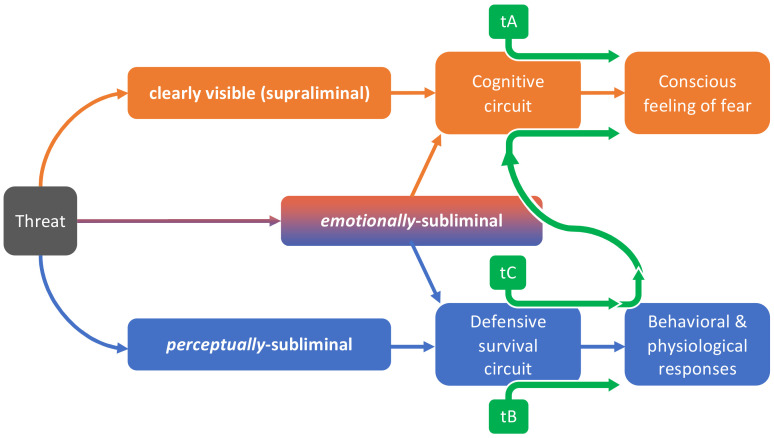
Integration of the bifactorial model previously proposed by Frumento and colleagues in 2021 [6] shows the impact of different levels of exposure to threats: while supraliminal and perceptually subliminal stimuli, respectively, impact the cognitive or the defensive survival circuits, emotionally-subliminal stimuli could impact on both circuits simultaneously. As a consequence, exposure therapies administering supraliminal stimuli only (tA) could leave behavioral and physiological correlates of fear dormant [23]; exposure therapies administering perceptually-subliminal stimuli only (tB) are ineffective in reducing the conscious feeling of fear; exposure therapies administering perceptually-subliminal, emotionally-subliminal and supraliminal stimuli in this order (tC) could maximize the benefit coming from each step.

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
