# Peer review of "Perceptually Visible but Emotionally Subliminal Stimuli to Improve Exposure Therapies"

_brainsci, 2022, doi:10.3390/brainsci12070867_

Round 1

Reviewer 1 Report

This paper offers a commentary on the possibilities offered by subliminal exposure to treat pathological fear, proposing that the use of perceptually-detectable but emotionally sublimininal feared stimuli has possibility as an intermediate step of exposure treatment. I find the authors idea of using such stimuli as a part of a graded exposure treatment intriguing, and the rationale - that emotionally subliminal stimuli would be non-aversive to patients while still targeting the subjective componenents of fear - to be logical. However, I'm not sure that an exclusively theoretical rationale for this idea, without presenting any empirical data that this is feasible or could be effective, is worthy at this point. Ultimately the authors are only raising hypotheses about the possible utility of emotionally subliminal stimuli as a clinical intervention and the specific impact this would have on defensive survival circuits, and to lay out these very testable hypotheses in published form without gathering any empirical data to test them seems premature. This seems particularly important given that there does not appear to be clear evidence yet that it is possible to deliberately and consistently present phobic stimuli that would be consciously perceptible but emotionally non-evocative to a given individual with a phobia. Rather, it seems the authors are inferring this possibility based on prior methodological challenges of the field in creating perceptually subliminal stimuli. Given how methodologically complicated this type of research is,  such feasibility is important first step to more clearly establish.

I also am not sure that the authors' claims that exposure to supraliminal stimuli "directly reduces subjective fear but does not necessarily suppress overreactivity of the defensive survival circuit..." (lines 160-162) and "could leave behavioral and physiological correlates of fear dormant" (Figure 2 caption) is well-supported by the literature. And it certainly needs to be better supported in the present article (the only citation provided is a popular press Scientific American article from 1994). While defensive threat responeses may not be entirely eliminated by exposure, this is true for subjective fear as well, and Im not aware of any research consistently showing improvements in subjective fear from exposure in the absence of corresponding improvements in common measures of defensive survival circtuiry (i.e. psychophysiology). Rather they typically go hand in hand, and a large body of literature shows exposure leads to improvements in both subjective and physiological/defensive survival measures (e.g., see Goncalves et al., 2015, Journal of Affective Disorders, or Diemer et al., 2014, World Journal of Biological Psychiatry). 

To reiterate my initial point, I think the ideas proposed in this are promising and worth further investigating, but think it would be best to have them paired with at least some data regarding the feasibility and preliminary efficacy of emotionally subliminal exposure before publication, and encourage the authors to collect such data if not already doing so. 

Reviewer 2 Report

This article brings attention to the crucial aspect of emotional subliminal stimuli and its prospective role in exposure therapies. The article is timely. I just have one suggestion: 

Line 71-72 the authors emphasized importance of assessing effectiveness of stimuli masking trial-by trial, which could interestingly be also monitored by trial by trial measures of brain responses to gather additional information on the effect of subliminal stimuli.  (See for instance: The Potential of Trial-by-Trial Variabilities of Ongoing-EEG, Evoked Potentials, Event Related Potentials and fMRI as Diagnostic Markers for Neuropsychiatric Disorders. Front. Neurosci. 12:850. doi: 10.3389/fnins.2018.00850)

Reviewer 3 Report

This Perspective article is very well written and provides and interesting and thought provocative distinction between perceptually- and emotionally-subliminal stimuli.

I don't have comments about the article itself, but I invite the authors to consider the potential importance of incorporating subliminal pictures presentation to other successful methods for exposure therapy, such as mental imagery (which is the primary tecnique used in exposure therapies, from phobia to PTSD). This integration could be particularly relevant if we consider recent evidences demonstrating that emotional imagery and emotional visual processing seems to activate distinct neural networks (PMID: 32011742), and exposing patients to  visual scenes may provide a more direct way to habituate to fear-relevant contents (PMID: 15670706) and "speed up" the therapy.

Author Response

This manuscript is a resubmission of an earlier submission. The following is a list of the peer review reports and author responses from that submission.